# Unsteady RANS Simulations of Strong and Weak 3D Stall Cells on a 2D Pitching Aerofoil

**Dajun Liu** [1,*] **and Takafumi Nishino** [2]

[1]   School of Water, Energy and Environment, Cranfield University, Bedfordshire MK43 0AL, UK
[2]   Department of Engineering Science, University of Oxford, Oxford OX1 3PJ, UK;
      takafumi.nishino@eng.ox.ac.uk
*   Correspondence: dajun.liu@cranfield.ac.uk

**Abstract:** A series of three-dimensional unsteady Reynolds-averaged Navier–Stokes (RANS) simulations are conducted to investigate the formation of stall cells over a pitching NACA 0012 aerofoil. Periodic boundary conditions are applied to the spanwise ends of the computational domain. Several different pitching ranges and frequencies are adopted. The influence of the pitching range and frequency on the lift coefficient ($C_L$) hysteresis loop and the development of leading-edge vortex (LEV) agrees with earlier studies in the literature. Depending on pitching range and frequency, the flow structures on the suction side of the aerofoil can be categorized into three types: (i) strong oscillatory stall cells resembling what are often observed on a static aerofoil; (ii) weak stall cells which are smaller in size and less oscillatory; and (iii) no stall cells at all (i.e., flow remains two-dimensional) or only very weak oval-shaped structures that have little impact on $C_L$. A clear difference in $C_L$ during the flow reattachment stage is observed between the cases with strong stall cells and with weak stall cells. For the cases with strong stall cells, arch-shaped flow structures are observed above the aerofoil. They resemble the $\Pi$-shaped vortices often observed over a pitching finite aspect ratio wing.

**Keywords:** pitching aerofoil; unsteady RANS; hysteresis loop; leading-edge vortex; stall cells

## 1. Introduction

Dynamic stall is a phenomenon that occurs when an aerofoil undergoes a rapid motion such as pitching, plunging, and flapping. In 1968, Ham and Garelick [1] performed experiments on a long-span, two-dimensional pitching NACA 0012 aerofoil and found that intense vorticity developed from the leading edge and shed from the aerofoil. This vortex influenced the aerodynamic loading and resulted in a larger stall angle and a higher lift coefficient than the static stall. A review by McCroskey [2] in 1981 summarizes that for a single aerofoil at a low Mach number, the three major parameters that can significantly affect the aerodynamic performance of dynamic stall are the reduced frequency, the mean angle, and the pitching amplitude (the latter two can be combined and described as the pitching range). Different reduced frequencies and pitching ranges result in different values of lift and drag coefficient ($C_L$ and $C_D$, respectively), stall angles, hysteresis effect, and flow regimes.

This influence has been investigated further by numerous studies. Brandon [3] performed experiments on a flat-plate wing and found that the overshoot of the maximum lift coefficient for a pitching aerofoil grew linearly with the reduced frequency within a certain range. A numerical study with a wider range of reduced frequencies but with different flow conditions by Akbari and Price [4] showed that the $C_L$ overshoot had a limit of 1.6, after which $C_L$ did not grow with the reduced frequency any more. An experimental research by Choudhry et al. [5] showed that the reduced frequency had a clear impact on the leading-edge vortex (LEV). A low reduced frequency may cause the disappearance of LEV, resulting in the lack of a sudden increase of $C_L$. An even lower reduced

frequency could eliminate the dynamic stall features completely and make the flow quasi-steady, which was also verified by the experimental study of Alrefai and Acharya [6]. However, Lee and Gerontakos [7] found in their experiments that the reduced frequency had little impact on the location and size of LEV. In the experimental and numerical study of Ducoin et al. [8], it was also found that the higher reduced frequency delayed the boundary layer transition and suppressed the transition-induced lift coefficient inflection. Higher reduced frequency resulted in a higher maximum lift value and a stronger hysteresis effect during the downstroke stage.

As for the pitching range, multiple researchers have found that a larger pitching amplitude produces a higher lift and a broadened hysteresis loop [2–6]. It was found by Brandon [3] that for a moderate pitching amplitude, an increasing mean pitch angle might transfer the flow from 'prestall' regime (flow remains attached) to 'deep stall' regime (flow separates and reattaches at a certain angle of attack (AoA)) and delay the reattachment to a higher AoA. On the other hand, for a fixed mean pitch angle, a smaller pitching amplitude tended to enlarge the hysteresis effect and made the stall onset happen during the downstroke stage. The numerical study of Akbari and Price [4] found that the separation of the flow happened earlier with an increased mean pitch angle.

The development of the LEV found in [2] has also been observed in other experimental [9] and numerical [10] studies. The general process can be described as follows: as the aerofoil is pitching up from a low AoA, a low-pressure region develops around the leading edge, which later turns into a leading-edge separation bubble (LSB). The bubble then develops into an LEV, travels along the upper surface of the aerofoil, and then sheds away from the trailing edge. It is believed that the onset of LEV is a major contribution to the enhancement of the aerodynamic loads [11]. As the LEV is shedding, new, smaller bubbles and vortices start to develop and shed from the surface, forming a vortex street in the wake. The shedding of the LEV, however, causes a sudden drop of the lift [9]. After the aerofoil begins to pitch down, the flow gradually reattaches onto the aerofoil upper surface.

The 3D effects of dynamic stall have also been the interest of researchers. In 1996, Wernert et al. [12] carried out wind tunnel experiments on a NACA 0012 aerofoil with a finite span. The particle image velocimetry and laser-sheet visualization indicated strong three-dimensional flow field during the downstroke stage. Three-dimensional spanwise topology was also observed by Disotell et al. [13] in the experiments on a Delft Du97-W-300 aerofoil. Fast-responding, pressure-sensitive paint and oil-flow visualization showed arch-shaped flow structures on the aerofoil surface near stall. The degree of spanwise nonuniformity decreased as the aerofoil pitched up. However, further pitching up caused a total three-dimensional breakdown of the flow field. The downstroke stage of dynamic stall was characterized by massive three-dimensionalities and strong pressure fluctuations. The three-dimensional effects are more frequently captured by numerical studies. Zanotti et al. [14,15] conducted experimental and numerical investigations on a pitching NACA 23012 aerofoil and detected strong spanwise three-dimensional structures at the downstroke stage. They claimed that it was due to the intrinsic three-dimensional nature of dynamic stall. However, Howell et al. [16] and Li et al. [17] attributed this phenomenon to the interaction between the LEV and the tip vortex. Drofelnik and Campobasso [18] stated that the three-dimensional effects were caused by the tip-vortex-induced downwash. Spentzos et al. [19,20] found out that the downstroke stage of dynamic stall was dominated by the interaction of a Ω-shaped vortex with the tip vortex. The two vortices merged and formed a Π–Ω configuration. More recently, Visbal and Garmann [21] adopted large eddy simulations (LES) and observed a Λ-shaped vortex passing away from the wing surface near the center line plane while remaining pinned at the corner during the downstroke stage.

The 3D effects of a stalled aerofoil have also been thoroughly studied with a static aerofoil. Moss and Murdin [22], Gregory et al. [23], Winkelmann et al. [24], and Winkelmann and Barlow [25] in their experiments observed counter-rotating vortices on the suction side of a static stalled aerofoil. These structures, which are usually called stall cells, can be unsteady under certain circumstances and have a significant impact on the lift coefficient [26]. The observation of stall cells under various flow conditions implies that the flow field of a stalled aerofoil is intrinsically three-dimensional, regardless

of the types of the spanwise end conditions. It is worth noticing that most of the experimental studies used a finite wing while some studies observed stall cell formation on very wide aerofoil (AR = 9–12) to eliminate the impact of the tip vortex. Recently, the authors of the present paper have also conducted a numerical study on this issue [27], further extending an earlier numerical study of Manni et al. [28]. In these studies, a NACA 0012 aerofoil with an infinitely long span at stall condition was investigated. The influence of the Reynolds number, aspect ratio (AR), and the spanwise resolution on the formation of 3D stall cells have been scrutinized.

Although 3D effects have been observed in various experimental and numerical studies on dynamic stall, their relevance to the stall cell formation on a static aerofoil has not been investigated in detail. In this study, we numerically investigate a NACA 0012 aerofoil under a sinusoidally pitching condition and the stall cell formation on the suction side. Different types of stall cells are observed and a comparison with the static stall cells is made.

## 2. Methodology

In this study, we focus on a NACA 0012 aerofoil pitching in a sinusoidal manner around its quarter-chord position (from the leading edge). The Reynolds number based on the chord length of the aerofoil is $1.35 \times 10^5$ (only for comparison with published experimental data) and $1 \times 10^6$ for the main cases. The Mach number of the flow is 0.006 and 0.043, respectively; therefore, the flow can be considered as incompressible. Unsteady RANS simulations are conducted with various reduced frequencies and pitching ranges. Their impact on the lift coefficient hysteresis loop is studied in detail. The behavior of the LEV as well as the formation of stall cells and other types of 3D flow structures on and above the aerofoil surface are also investigated.

### 2.1. Governing Equations

The incompressible RANS equations can be written in Cartesian tensor form as:

$$\frac{\partial U_i}{\partial x_i} = 0 \tag{1}$$

$$\rho \frac{\partial U_i}{\partial t} + \rho \frac{\partial}{\partial x_j}(U_i U_j) = -\frac{\partial P}{\partial x_i} + \frac{\partial}{\partial x_j}[\mu(\frac{\partial U_i}{\partial x_j} + \frac{\partial U_j}{\partial x_i}) - \rho \overline{u_i' u_j'}] \tag{2}$$

Under the Boussinesq hypothesis, the Reynolds stress tensor $-\rho \overline{u_i' u_j'}$ can be expressed as:

$$-\overline{\rho u_i' u_j'} = \mu_t \left(\frac{\partial U_i}{\partial x_j} + \frac{\partial U_j}{\partial x_i}\right) - \frac{2}{3}\rho k \delta_{ij} \tag{3}$$

where $\mu_t$ is the turbulence eddy viscosity, $k$ is the turbulent kinetic energy, and $\delta_{ij}$ is the Kronecker delta.

The SST $k$-$\omega$ turbulence model [29] is employed to compute the eddy viscosity. The choice of the turbulence model is based on its successful application in CFD simulations on a pitching aerofoil. Its accuracy has been shown to be acceptable in predicting the maximum lift coefficient and stall angle [30–33]. Eddy-solving simulations such as DES and LES would yield more accurate results. However, the computational cost would be too large for the current study. Therefore, SST $k$-$\omega$ model is chosen to achieve a balance between accuracy and computation cost.

In the SST $k$-$\omega$ model, the eddy viscosity is calculated through solving two variables: the turbulent kinetic energy $k$ and its specific dissipation rate $\omega$. They are obtained from the following transport equations [34]:

$$\rho \frac{\partial k}{\partial t} + U_i \frac{\partial k}{\partial x_i} = \frac{\partial}{\partial x_j}\left(\Gamma_k \frac{\partial k}{\partial x_j}\right) + G_k - Y_k \tag{4}$$

$$\rho\frac{\partial\omega}{\partial t} + U_i\frac{\partial\omega}{\partial x_i} = \frac{\partial}{\partial x_j}\left(\Gamma_\omega\frac{\partial\omega}{\partial x_j}\right) + G_\omega - Y_\omega + D_\omega \tag{5}$$

where $G_k$ and $G_\omega$ represent the generation of $k$ and $\omega$, respectively. $Y_k$ and $Y_\omega$ represent the dissipation of $k$ and $\omega$, respectively, due to turbulence. $\Gamma_k$ and $\Gamma_\omega$ represent the effective diffusivity of $k$ and $\omega$, respectively. $D_\omega$ represents the cross-diffusion term. They are computed as follows:

$$G_k = \mu_t S^2 \tag{6}$$

where $S \equiv \sqrt{2S_{ij}S_{ij}}$ and $S_{ij} = \frac{1}{2}\left(\frac{\partial U_i}{\partial x_j} + \frac{\partial U_j}{\partial x_i}\right)$.

$$G_\omega = \frac{\rho\alpha\alpha^*}{\mu_t}G_k \tag{7}$$

where $\alpha = \frac{\alpha_\infty}{\alpha^*}\left(\frac{\alpha_0 + \frac{Re_t}{R_\omega}}{1 + \frac{Re_t}{R_\omega}}\right)$, $\alpha^* = \frac{\alpha_0^* + \frac{Re_t}{R_k}}{1 + \frac{Re_t}{R_k}}$, $\alpha_\infty = F_1\left(\frac{\beta_{i,1}}{\beta_\infty^*} - \frac{\kappa^2}{\sigma_{\omega,1}\sqrt{\beta_\infty^*}}\right) + (1 - F_1)\left(\frac{\beta_{i,2}}{\beta_\infty^*} - \frac{\kappa^2}{\sigma_{\omega,2}\sqrt{\beta_\infty^*}}\right)$,
$F_1 = \tanh\left(\Phi_1^4\right)$, $\Phi_1 = \min\left[\max\left(\frac{\sqrt{k}}{0.09\omega y}, \frac{500\mu}{\rho y^2\omega}\right), \frac{4\rho k}{\sigma_{\omega,2}D_\omega^+ y^2}\right]$, $Re_t = \frac{\rho k}{\mu\omega}$, $R_\omega = 2.95$, $R_k = 6$, $\alpha_0 = \frac{1}{9}$,
$\alpha_0^* = 0.024$, $\kappa = 0.41$, $\sigma_{\omega,1} = 2.0$, $\sigma_{\omega,2} = 1.168$, $\beta_\infty^* = 0.09$, $\beta_{i,1} = 0.075$, and $\beta_{i,2} = 0.0828$. In addition,
$y$ is the distance to the nearest wall surface, $\mu$ is the molecular viscosity of air, and $D_\omega^+$ is the positive portion of the cross-diffusion term $D_\omega$ which is calculated as: $D_\omega^+ = \max\left[2\rho\frac{1}{\sigma_{\omega,2}}\frac{1}{\omega}\frac{\partial k}{\partial x_j}\frac{\partial\omega}{\partial x_j}, 10^{-10}\right]$,

$$Y_k = \rho\beta^* k\omega \tag{8}$$

where $\beta^* = \beta_\infty^*\left(\frac{\frac{4}{15} + \left(\frac{Re_t}{R_\beta}\right)^4}{1 + \left(\frac{Re_t}{R_\beta}\right)^4}\right)$, $R_\beta = 8$.

$$Y_\omega = \rho\beta\omega^2 \tag{9}$$

where $\beta = F_1\beta_{i,1} + (1 - F_1)\beta_{i,2}$,

$$D_\omega = 2(1 - F_1)\rho\frac{1}{\omega\sigma_{\omega,2}}\frac{\partial k}{\partial x_j}\frac{\partial\omega}{\partial x_j} \tag{10}$$

$$\Gamma_k = \mu + \frac{\mu_t}{\sigma_k} \tag{11}$$

$$\Gamma_\omega = \mu + \frac{\mu_t}{\sigma_\omega} \tag{12}$$

where $\sigma_k$ and $\sigma_\omega$ are the turbulent Prandtl numbers for $k$ and $\omega$, respectively, calculated as $\sigma_k = \frac{1}{\frac{F_1}{\sigma_{k,1}} + \frac{1-F_1}{\sigma_{k,2}}}$, $\sigma_\omega = \frac{1}{\frac{F_1}{\sigma_{\omega,1}} + \frac{1-F_1}{\sigma_{\omega,2}}}$, where $\sigma_{k,1} = 1.176$, $\sigma_{k,2} = 1.0$.

Finally, the turbulent eddy viscosity $\mu_t$ is calculated as:

$$\mu_t = \frac{\rho k}{\omega}\frac{1}{\max\left[\frac{1}{\alpha^*}, \frac{SF_2}{a_1\omega}\right]} \tag{13}$$

where $F_2 = \tanh\left(\Phi_2^2\right)$, $\Phi_2 = \max\left[2\frac{\sqrt{k}}{0.09\omega y}, \frac{500\mu}{\rho y^2\omega}\right]$, and $a_1 = 0.31$.

### 2.2. Flow Configuration

The pitching motion of the aerofoil is defined by

$$\alpha(t) = \alpha_m + 0.5\alpha_1\sin(\Omega t) \tag{14}$$

where $\alpha(t)$ is the instantaneous AoA, $\alpha_m$ is the mean angle, $\alpha_1$ is the pitching range, and $\Omega$ is the angular velocity. In this study, multiple sets of $\alpha_m$ and $\alpha_1$ are adopted to investigate the impact of mean angle and pitching range, respectively. For all cases, the minimum AoA is fixed at $-5°$ and the maximum AoA varies from $25°$ down to a prestall angle. The choice of these parameters is adopted from Lee and Gerontakos [7] for comparison.

The pitching reduced frequency $f$ is defined by:

$$f = \frac{\Omega c}{2U_\infty} \tag{15}$$

where $c$ is the aerofoil chord length, which is 1m in this study, and $U_\infty$ is the free-stream velocity. Three different reduced frequencies (0.01, 0.025, and 0.1) are tested to investigate the influence of the pitching rate. The choice of the reduced frequencies is loosely based on the practical flow conditions of both commercial and research vertical axis wind turbines (VAWT) [35–39], where the dynamic stall may constantly occur to the turbine blades and the reduced frequency ranges from 0.01 to 0.1. A list of pitching angles and reduced frequencies of all cases in this study is shown in Table 1.

**Table 1.** Pitching angles and reduced frequencies investigated.

| Reduced Frequency $f$ | Mean Pitching Angle $\alpha_0$ (°) | Pitching Range $\alpha_1$ (°) | Maximum Angle $\alpha_0 + 0.5\alpha_1$ (°) | Minimum Angle $\alpha_0 - 0.5\alpha_1$ (°) |
|---|---|---|---|---|
| 0.01 | 6, 6.5, 7, ... 10 | 22, 23, 24, ... 30 | 17, 18, 19, ... 25 | −5 |
| 0.025 | 6.5, 7, 7.5, ... 10 | 23, 24, 25, ... 30 | 18, 19, 20, ... 25 | −5 |
| 0.1 | 7.5, 8, 8.5, ... 10 | 25, 26, 27, ... 30 | 20, 21, 22, ... 25 | −5 |

### 2.3. Computational Mesh

The computational mesh used in this study consists of two parts. Figure 1 shows a rotating O-mesh around the aerofoil with a 5 m radius and a static O-mesh as the outer region with a 5 m inner radius and a 50 m outer radius. For the rotating mesh, 436 nodes are arranged on the 2D aerofoil profile and 72 nodes are allocated in the radial direction. The first node distance from the aerofoil surface is set at $1 \times 10^{-4}$ m for $Re = 1.35 \times 10^5$ and $8 \times 10^{-6}$ m for $Re = 1 \times 10^6$ to make sure $y^+ < 1$ for each Reynolds number case. For the static mesh, 436 nodes and 83 nodes are allocated in the circumferential and radial directions, respectively, and the height of the first layer is set at the same value as the outer edge layer of the rotating mesh. The spanwise extent of the computational domain is 2.5$c$ and the number of layers is 25 (i.e., the spanwise resolution is 10% of the chord). This level of spanwise resolution has been shown to be able to capture the unsteadiness of stall cells over a static aerofoil [27].

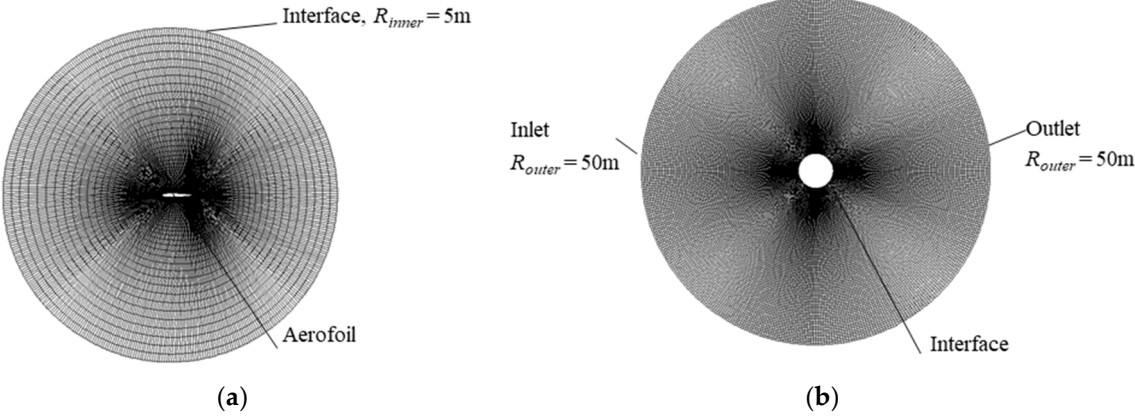

(a)　　　　　　　　　　　　　　　　　　　　　(b)

**Figure 1.** The rotating (inner) mesh and the static (outer) mesh. (**a**) Rotating mesh; (**b**) static mesh.

### 2.4. Computational Details

Simulations are performed using a commercial finite-volume solver ANSYS FLUENT 18.2. Periodic boundary conditions are applied to the spanwise direction. The air density is 1.225 kg/m$^3$ and the viscosity is $1.7894 \times 10^{-5}$ kg/(m·s). The free-stream velocity $U_\infty$ given at the inlet is 1.97 m/s for chord Reynolds number $Re = 1.35 \times 10^5$ (only for validation cases) and 14.607 m/s for $Re = 1 \times 10^6$. The second-order upwind scheme is used for spatial discretization and a second-order implicit scheme for time discretization. The SIMPLE algorithm [40] is adopted for the pressure–velocity coupling. At the outlet, the 'pressure outlet' condition (enforcing zero gradients for all velocity components and turbulence quantities) is employed with a fixed gauge pressure of 0 Pa. The turbulence intensity and viscosity ratio at the inlet are set at 1% and 1, respectively, to simulate a low free-stream turbulence level (a turbulent kinetic energy of $3.20 \times 10^{-2}$ m$^2$/s$^2$ and a specific dissipation rate of $2.19 \times 10^3$ s$^{-1}$).

The convergence criterion is set as $10^{-6}$ for the residuals of all flow variables solved at each time step. The flow time simulated is 86 s for $f = 0.01$, 34.4 s for $f = 0.025$, and 8.6 s for $f = 0.1$ to obtain four pitching cycles for each case. For all $f = 0.1$ cases and $f = 0.025$ cases with maximum AoA = 22°, 23°, 24°, and 25°, the results are fully converged (i.e., the $C_L$ hysteresis loops obtained from the third and fourth pitching cycles are identical). However, for the other cases, the $C_L$ hysteresis loop does not converge at a certain stage of dynamic stall. Figure 2 shows an example of $f = 0.01$, maximum AoA = 24°. When the aerofoil is pitching down from around 19° to 10° (represented in the following text as 19°↓ to 10°↓), the $C_L$ of the second, third, and fourth cycle does not agree. This is due to the formation of 3D asymmetric stall cells described later in this paper; their quantitative features are similar from cycle to cycle but they do not keep the exact same profile. In the following sections, unless stated otherwise, we investigate only the fourth cycle as a representative cycle.

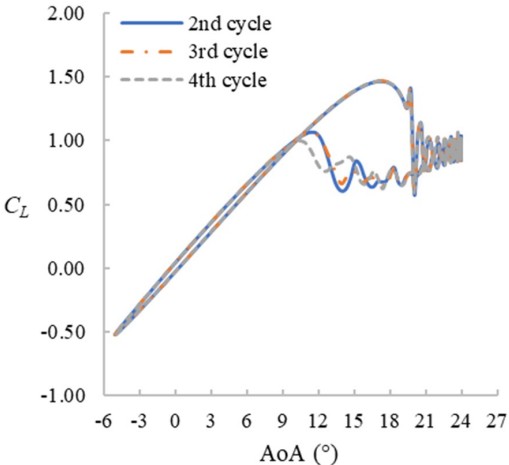

**Figure 2.** $C_L$ hysteresis loop for the second, third, and fourth cycles. $f = 0.01$, maximum AoA = 24°.

## 3. Sensitivity Analysis

The sensitivity to the mesh resolution and time step size is analyzed using a series of 2D simulations (i.e., using only one lateral layer of the 3D mesh described earlier in Section 2.2) at both static and pitching conditions. For the static condition, the Reynolds number is set as $Re = 1 \times 10^6$, AoA = 1°–19°, while the pitching case is set as $Re = 1.35 \times 10^5, f = 0.1$, and $\alpha(t) = 10° + 15° \sin(\omega t)$. The original mesh (G2, 436 × 155) is coarsened once with half the total nodes (G1, 308 × 110) and refined once with double the total nodes (G3, 616 × 220) while keeping the same first node distance from the aerofoil surface. Three different time step sizes, $\Delta t = 0.002$s, 0.001s, and 0.0005s, are also tested with the original mesh (G2). Figure 3 shows the mesh convergence under static condition at $Re = 1 \times 10^6$. As the figure suggests, the meshes for G2, G3, and G4 yield very similar results. Therefore, G2 can be considered as the lowest resolution to achieve mesh convergence. Figures 4 and 5 compare results for the pitching case with the experimental results of Lee and Gerontakos [7]. As the figures

show, the simulation underpredicts the stall angle and the maximum lift coefficient. However, the downstroke stage from 18°↓ to 9°↓ for G2 and G3 gives a fairly good prediction. In the next sections, we will find out that this AoA range is what this study actually focuses on. Also, for stalling and downstroke stages, the results do not improve substantially by increasing the mesh resolution or by reducing the time step size. Therefore, considering the computational cost, the mesh G2 and time step size of 0.001s are adopted for the rest of the study.

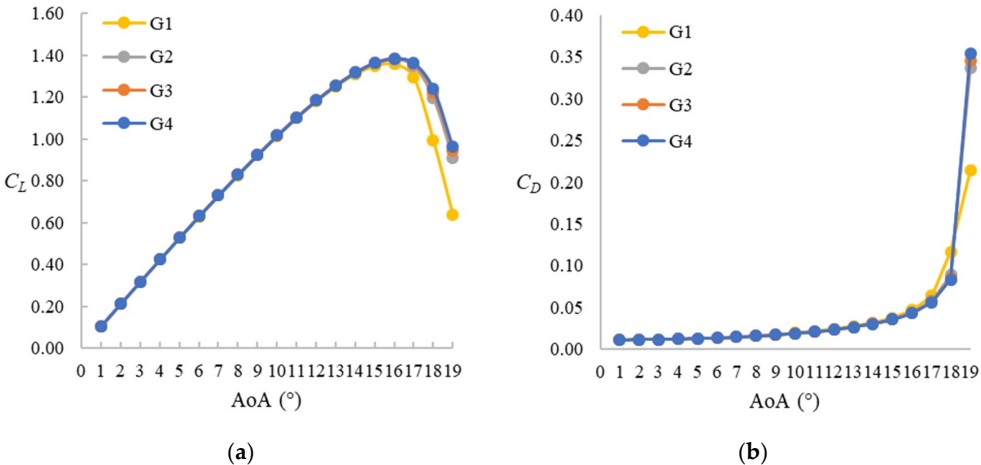

(**a**)　　　　　　　　　　　　　　　　　　　(**b**)

**Figure 3.** Lift and drag coefficient for different mesh resolutions (2D, Δt = 0.001 s). Static stall.

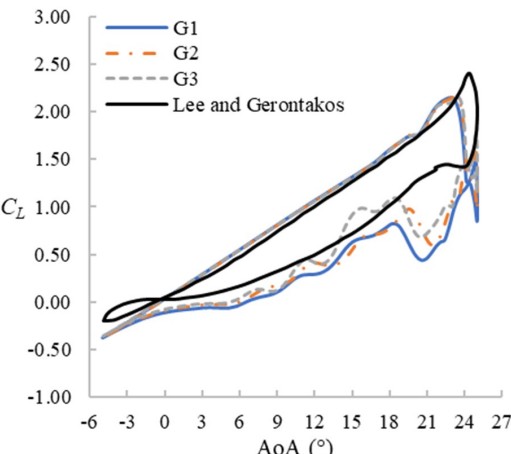

**Figure 4.** Lift hysteresis loop for different mesh resolutions (2D, Δt = 0.001 s).

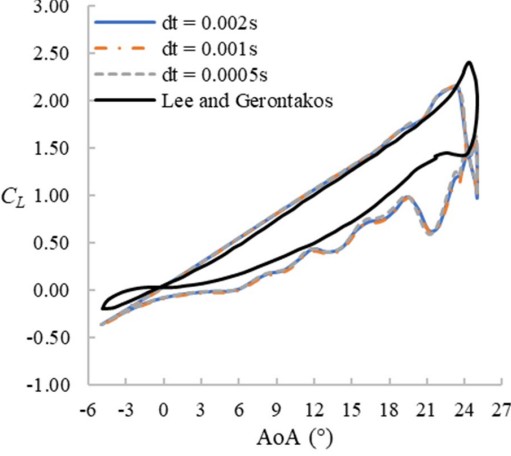

**Figure 5.** Lift hysteresis loop for different time step sizes (2D, original mesh G2).

## 4. Results

*4.1. Cases with f = 0.1*

### 4.1.1. Lift Coefficient ($C_L$) Hysteresis Loop

The $C_L$ hysteresis loops for $f$ = 0.1 cases are presented in Figure 6. The maximum AoA ranges from 20° to 25° with the 20° case being the prestall case. For cases with higher maximum AoA, the stall happens later. During the upstroke stage up to about 18°, the $C_L$ value is not affected by the pitching range and varies almost linearly with AoA. The deep-stall cases share some similar features: a sudden increase of $C_L$ followed by a significant drop. Except for the case with the widest pitching range (Max AoA = 25°) and the prestall case (Max AoA = 20°), the sudden increase of $C_L$ occurs at the beginning of the downstroke stage (i.e., after (not before) reaching the maximum AoA). The $C_L$ variation during the downstroke stage of the pitching is a little chaotic but similar fluctuation patterns can be seen for all cases except for the prestall case. The hysteresis effect is very obvious in all cases.

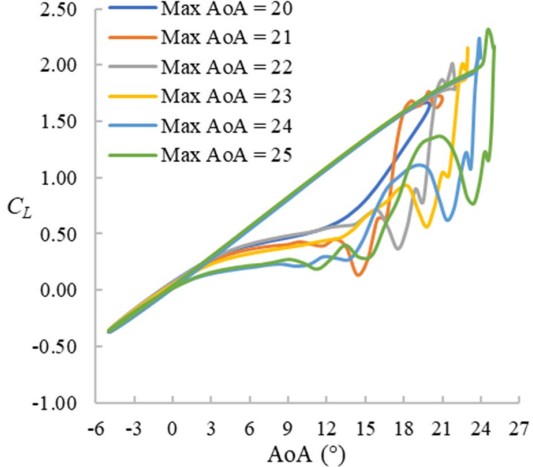

**Figure 6.** $C_L$ hysteresis loops for $f$ = 0.1 cases.

### 4.1.2. Leading-Edge Vortex (LEV) and Trailing-Edge Separation

The dynamic stall process is visualized by a series of 2D snapshots of pressure field and streamlines around the aerofoil at the midspan position for the maximum AoA = 23° case in Figure 7. First, a separation bubble develops from the trailing edge up to near the leading edge (Figure 7a). Then, the flow separates from the leading edge and a leading-edge vortex (LEV) is formed, which gradually develops towards the trailing edge whilst the size of the original separation bubble (developed from the trailing edge) gradually decreases (Figure 7b). The fully developed LEV separates from the leading edge (Figure 7c) and is then detached from the surface of the aerofoil whilst a new small vortex starts to develop from the trailing edge (Figure 7d). This new vortex develops above the trailing edge whilst the LEV travels downstream (Figure 7e, f); note that the new vortex from the trailing edge is an anticlockwise vortex whereas the LEV is a clockwise vortex. Eventually, this trailing-edge vortex also sheds from the aerofoil (Figure 7g). No further vortex shedding is observed as the AoA is already lower than the (static) stall angle at this stage (Figure 7h) but the flow remains to be separated until the AoA becomes much lower. The sudden increase of $C_L$ corresponds to the early development of LEV (Figure 7b), whereas the sudden drop corresponds to the shedding of the LEV (Figure 7d). A similar dynamic stall process has been observed in all cases with $f$ = 0.1 (except for the prestall case).

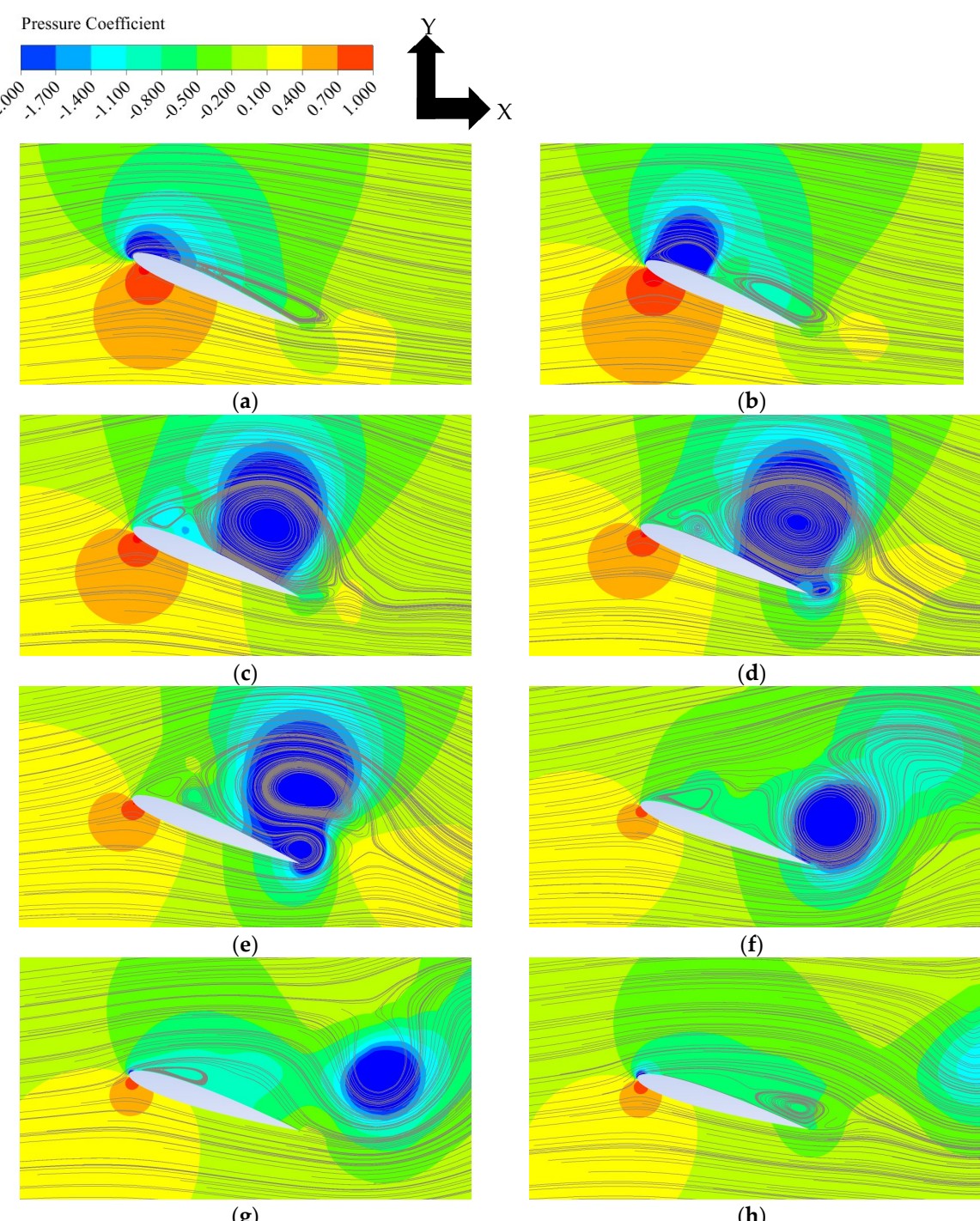

**Figure 7.** Pressure field and stream lines around the aerofoil. Maximum AoA = 23°, *f* = 0.1.
(**a**) AoA = 22.90°↑; (**b**) AoA = 22.99°↓; (**c**) AoA = 22.28°↓; (**d**) AoA = 21.92°↓; (**e**) AoA = 21.50°↓;
(**f**) AoA = 19.83°↓; (**g**) AoA = 18.42°↓; (**h**) AoA = 16.81°↓.

### 4.1.3. 3D Flow Structures

3D flow structures on the aerofoil surface are observed only in the case with the maximum AoA = 24° during the downstroke stage, as shown by streamlines colored by streamwise wall shear stress in Figure 8. Since the aerofoil is pitching, a local coordinate system fixed to the aerofoil is adopted in this figure and the following figures showing the aerofoil surface. As shown, some small 3D structures develop near the leading edge and transform into oval-shaped structures and then move toward the trailing edge. These 3D structures are very weak and have little impact on the lift coefficient. In all other cases with $f$ = 0.1, the flow field above the aerofoil surface remains two-dimensional.

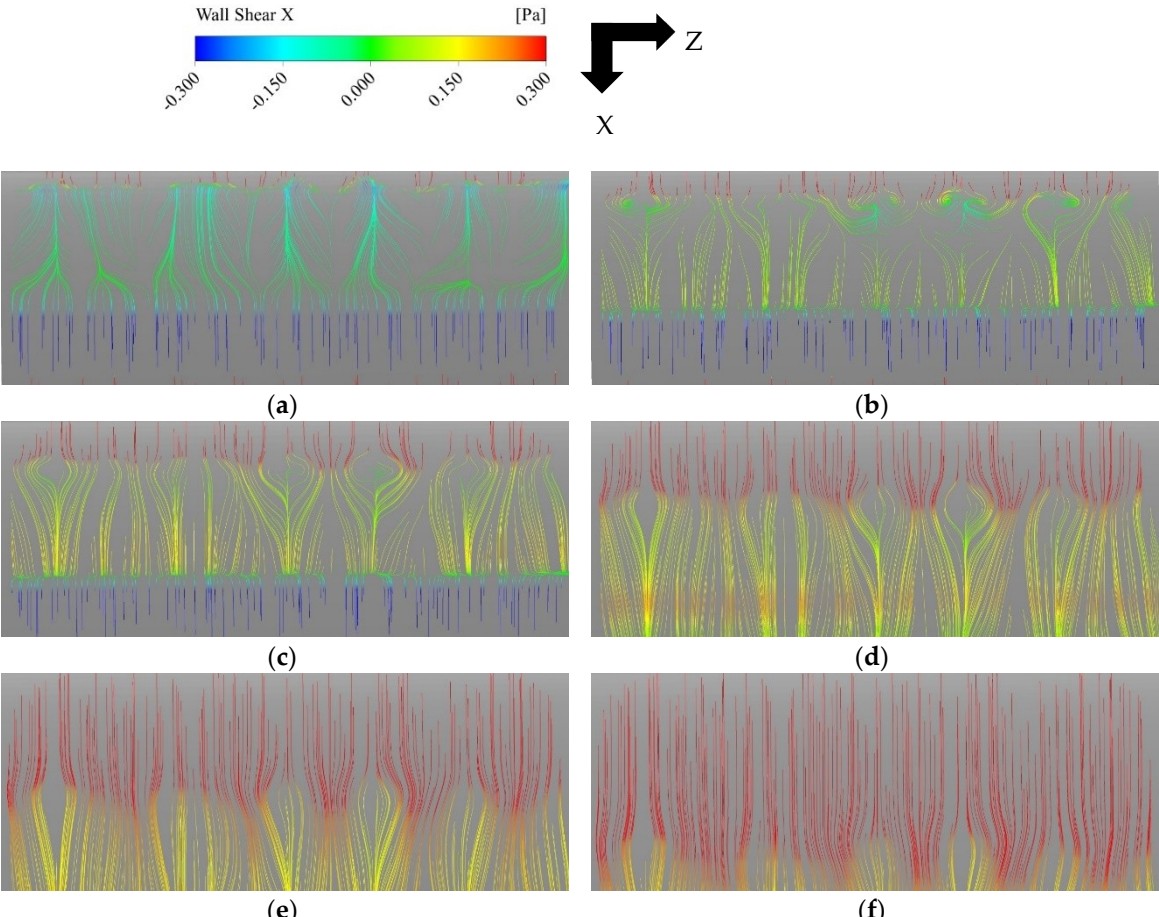

**Figure 8.** 3D flow structures on the aerofoil surface. Maximum AoA = 24°, $f$ = 0.1. (**a**) AoA = 13.81°↓; (**b**) AoA = 12.79°↓; (**c**) AoA = 11.75°↓; (**d**) AoA = 9.64°↓; (**e**) AoA = 7.53°↓; (**f**) AoA = 5.46°↓.

### 4.2. Cases with f = 0.025

#### 4.2.1. $C_L$ Hysteresis Loop

The maximum AoA tested for the cases with $f$ = 0.025 ranges from 18° to 25°. The case with maximum AoA = 18° is a prestall case. The $C_L$ hysteresis loops of five selected cases are presented in Figure 9. They have some similarities to those for the $f$ = 0.1 cases presented earlier; the loops start with a linearly increasing $C_L$ followed by a sudden drop. However, in these cases with $f$ = 0.025, the stall starts before reaching the maximum AoA, and after the drop, the $C_L$ fluctuates periodically until around 17°↓.

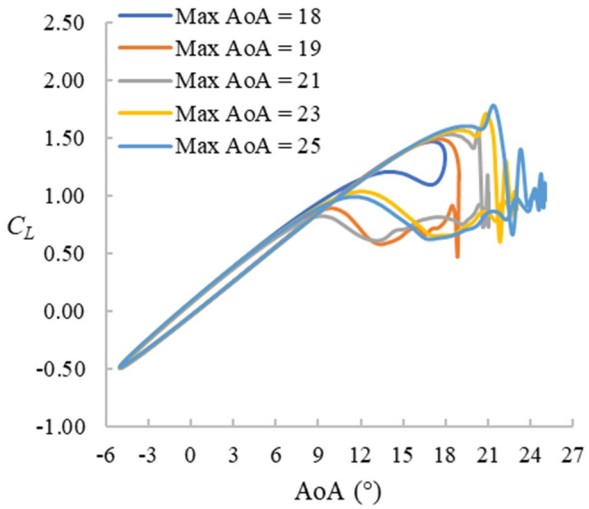

**Figure 9.** $C_L$ hysteresis loops for $f$ = 0.025 cases.

### 4.2.2. LEV and Trailing-Edge Separation

The developments of LEV and trailing-edge separation are similar to those for the $f$ = 0.1 cases. However, due to the early onset of stall and the lower pitching frequency, several sets of LEV and trailing-edge vortices are formed and shed from the aerofoil during the late upstroke stage and early downstroke stage, causing the periodic fluctuation of the $C_L$.

### 4.2.3. Stall Cell Formation

For $f$ = 0.025, 3D flow structures on the aerofoil surface are observed in all deep-stall cases. They can be categorized into two different types. For maximum AoA = 19°, 20°, and 21°, the 3D structures resemble the stall cell structures often observed for a static aerofoil, such as those in [27,28]. For these cases, small 3D structures start to develop near the leading edge when the aerofoil is pitching down to around 17° and then merge into large counter-rotating vortex pairs dominating the whole span. As the aerofoil further pitches down, these stall cells move toward the trailing edge and shed from the aerofoil. The surface flow pattern then becomes 2D again. The maximum AoA = 21° case is shown in Figure 10 as an example. The other two cases show very similar characteristics.

For maximum AoA = 22°, 23°, 24°, and 25°, much smaller counter-rotating vortices start to develop around a quarter chord away from the leading edge. Then, they move toward the trailing edge with their size almost unchanged. Compared to the very weak 3D structures observed earlier in $f$ = 0.1 case, these structures have a clearer stall-cell-like shape. However, they are smaller and weaker compared to the stall cells observed in maximum AoA = 19°, 20°, and 21° cases with the same reduced frequency. Therefore, we shall call these structures 'weak stall cells' in the following discussion. The maximum AoA = 23° case is shown in Figure 11 as an example. The other three cases show very similar characteristics.

Importantly, the difference in the strength of stall cells also has an impact on the lift coefficient, as shown earlier in Figure 10. For the cases with weak stall cells, the lift gradually recovers from 17°↓ to 13°↓, which is the range of AoA that allows stall cells to develop. For the cases with strong stall cells, however, the lift does not recover during this AoA range as the reattachment of the flow is delayed compared to the cases with weak stall cells.

Judging from the $C_L$ hysteresis loop, stall cells, strong or weak, always start to develop after the periodic forming of LEVs and trailing-edge vortices (corresponding to the periodic oscillation of $C_L$).

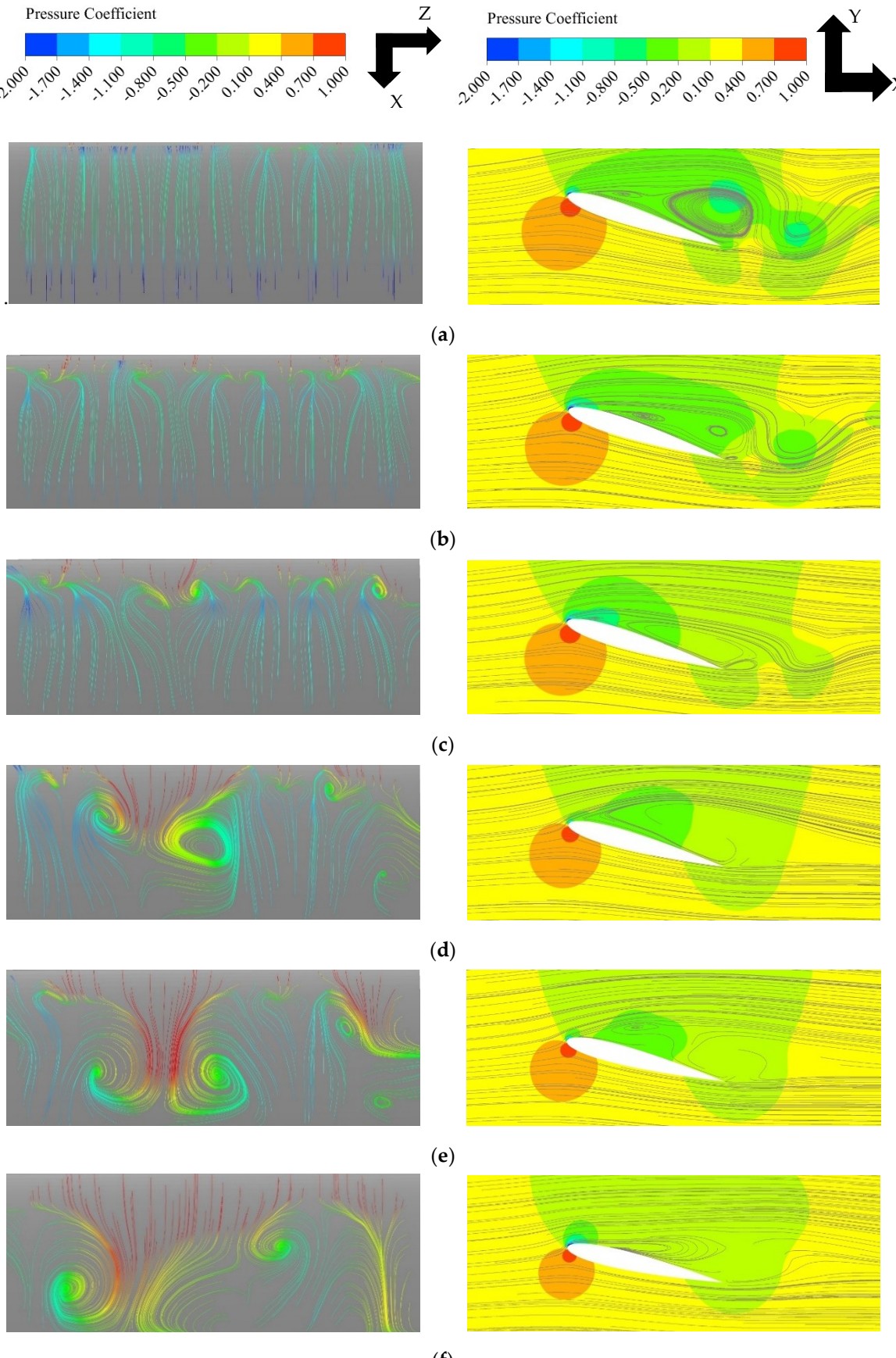

**Figure 10.** *Cont.*

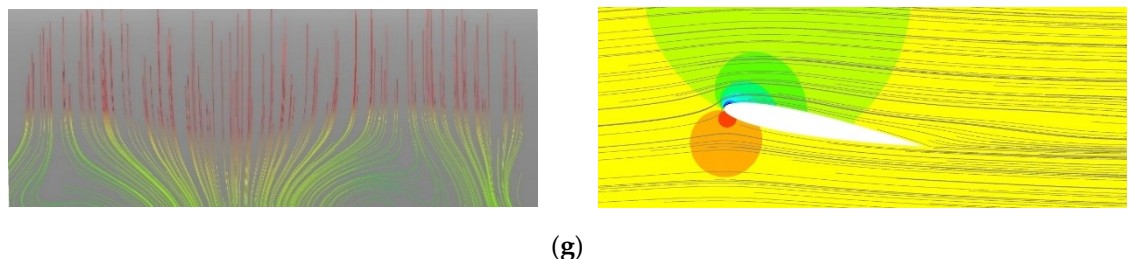

(**g**)

**Figure 10.** Stall cells on the aerofoil surface. Maximum AoA = 21°, $f$ = 0.025. (**a**) AoA = 18.01°↓;
(**b**) AoA = 17.38°↓; (**c**) AoA = 16.69°↓; (**d**) AoA = 15.20°↓; (**e**) AoA = 14.39°↓; (**f**) AoA = 12.67°↓;
(**g**) AoA = 10.86°↓.

(**a**)

(**b**)

(**c**)

(**d**)

**Figure 11.** *Cont.*

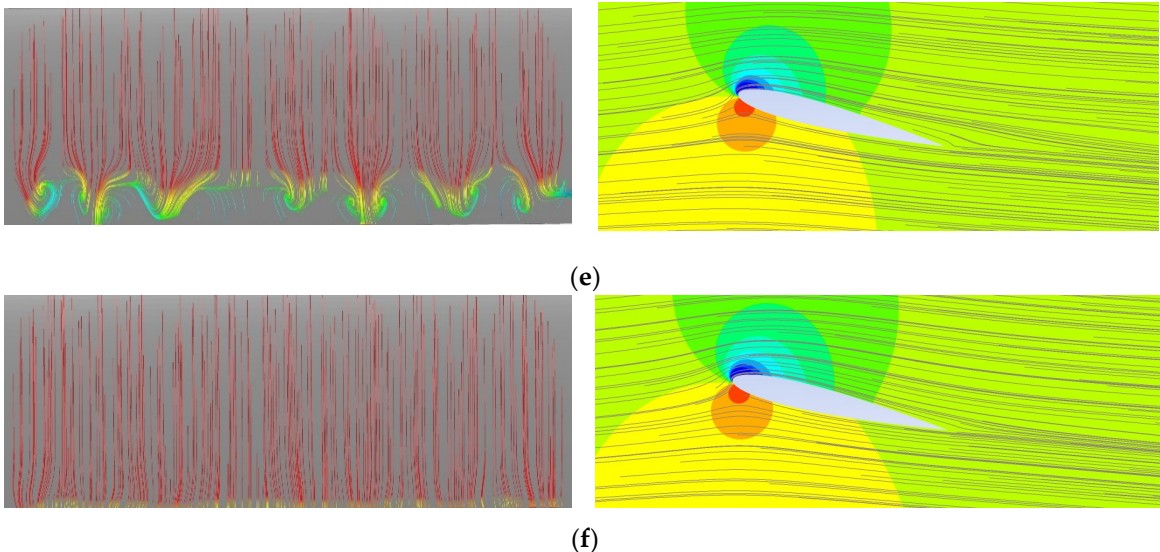

(e)

(f)

**Figure 11.** Weak stall cells on the aerofoil surface. Maximum AoA = 23°, $f$ = 0.025. (**a**) AoA = 17.69°↓; (**b**) AoA = 16.87°↓; (**c**) AoA = 16.00°↓; (**d**) AoA = 15.10°↓; (**e**) AoA = 14.17°↓; (**f**) AoA = 13.20°↓.

*4.3. Cases with f = 0.01*

4.3.1. $C_L$ Hysteresis Loop

The maximum AoA for the cases with $f$ = 0.01 ranges from 17° to 25°. The case with maximum AoA = 17° is a prestall case. The $C_L$ hysteresis loops of four selected cases are presented in Figure 12. The $C_L$ hysteresis loops are very similar to those for the $f$ = 0.025 cases except that the stall takes place even earlier.

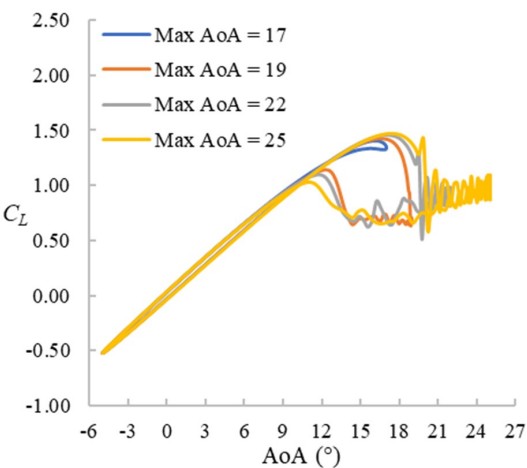

**Figure 12.** $C_L$ hysteresis loops for $f$ = 0.01 cases.

4.3.2. LEV and Trailing-Edge Separation

The process is visualized by a series of snapshots of pressure field and the stream lines around the aerofoil for maximum AoA = 21° case in Figure 13. Compared with the $f$ = 0.1 and 0.025 cases, the trailing-edge separation develops further before a LEV is created (Figure 13a). A weak LEV is then created (Figure 13b), which then grows and merges with the trailing-edge separation into a big clockwise vortex and dominates over the aerofoil (Figure 13c). The vortex then moves towards the trailing edge whilst a secondary counterclockwise vortex grows around the trailing edge (Figure 13d). The secondary vortex grows in size and the original clockwise vortex shrinks until they become approximately the same size (Figure 13e). Then, the secondary vortex sheds from the trailing edge

(Figure 13f) and another counterclockwise rotating vortex develops near the trailing edge and grows (Figure 13g, h). The trailing-edge vortex keeps developing and shedding, while the main clockwise vortex stays above the aerofoil and keeps expanding and shrinking in size. This process continues for several periods, corresponding to the $C_L$ fluctuation after stall.

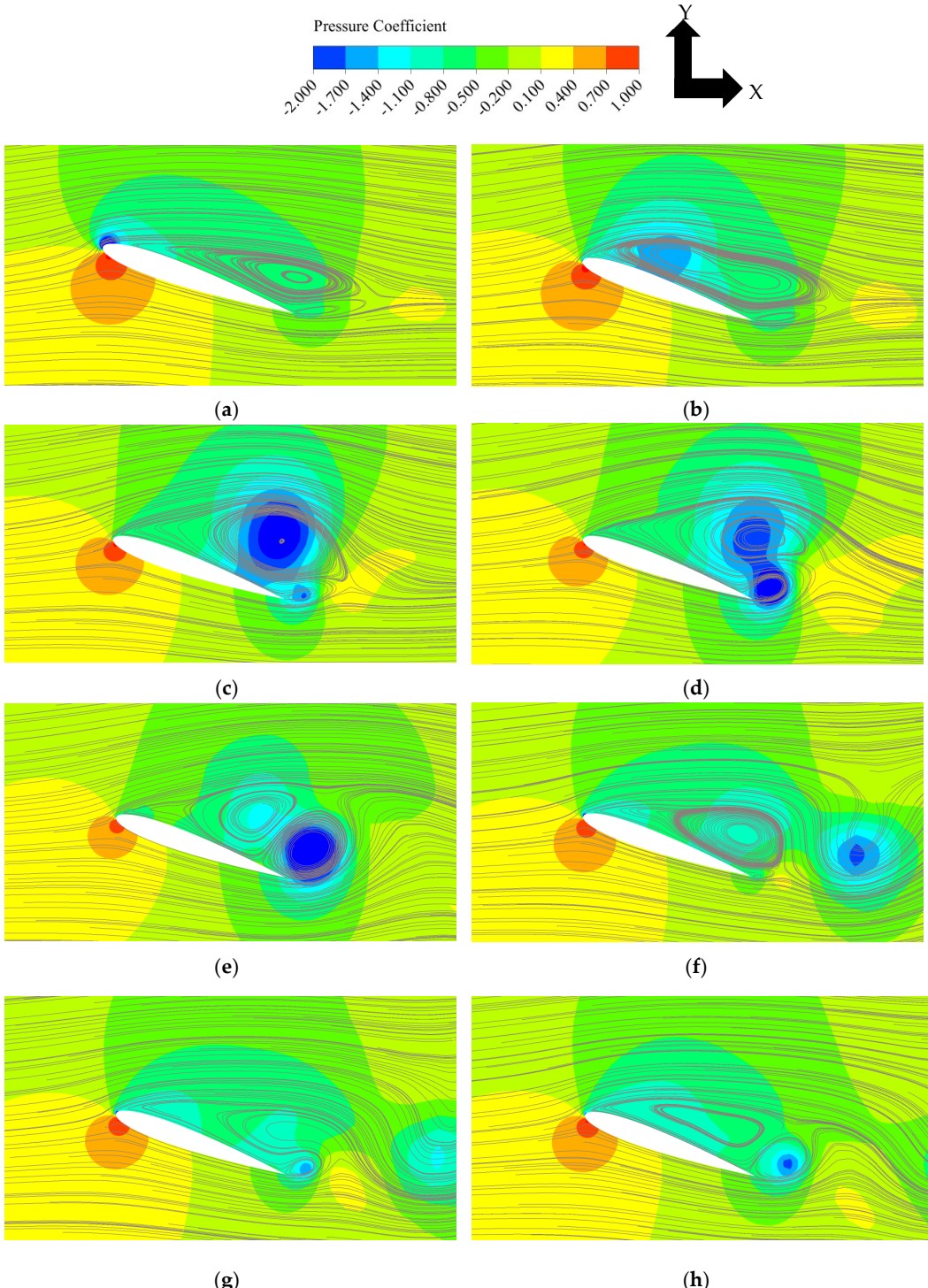

**Figure 13.** Pressure field and stream lines around the aerofoil. Maximum AoA = 21°, *f* = 0.01.
(**a**) AoA = 19.13°↑; (**b**) AoA = 19.27°↑; (**c**) AoA = 19.41°↑; (**d**) AoA = 19.46°↑; (**e**) AoA = 19.55°↑;
(**f**) AoA = 19.67°↑; (**g**) AoA = 19.72°↑; (**h**) AoA = 19.76°↑.

### 4.3.3. Stall Cell Formation

For $f$ = 0.01, strong and clear stall cell structures are observed in all deep-stall cases. The maximum AoA = 21° case is shown in Figure 14 as an example. When the aerofoil is pitching down to around 19°, small perturbations develop near the leading edge and then merge into counter-rotating vortex pairs. As the aerofoil pitches down further, the stall cells grow in size, move toward the trailing edge, and eventually shed from the aerofoil. The surface flow then becomes 2D again. All other cases except for the maximum AoA = 17° (prestall case) show similar flow characteristics.

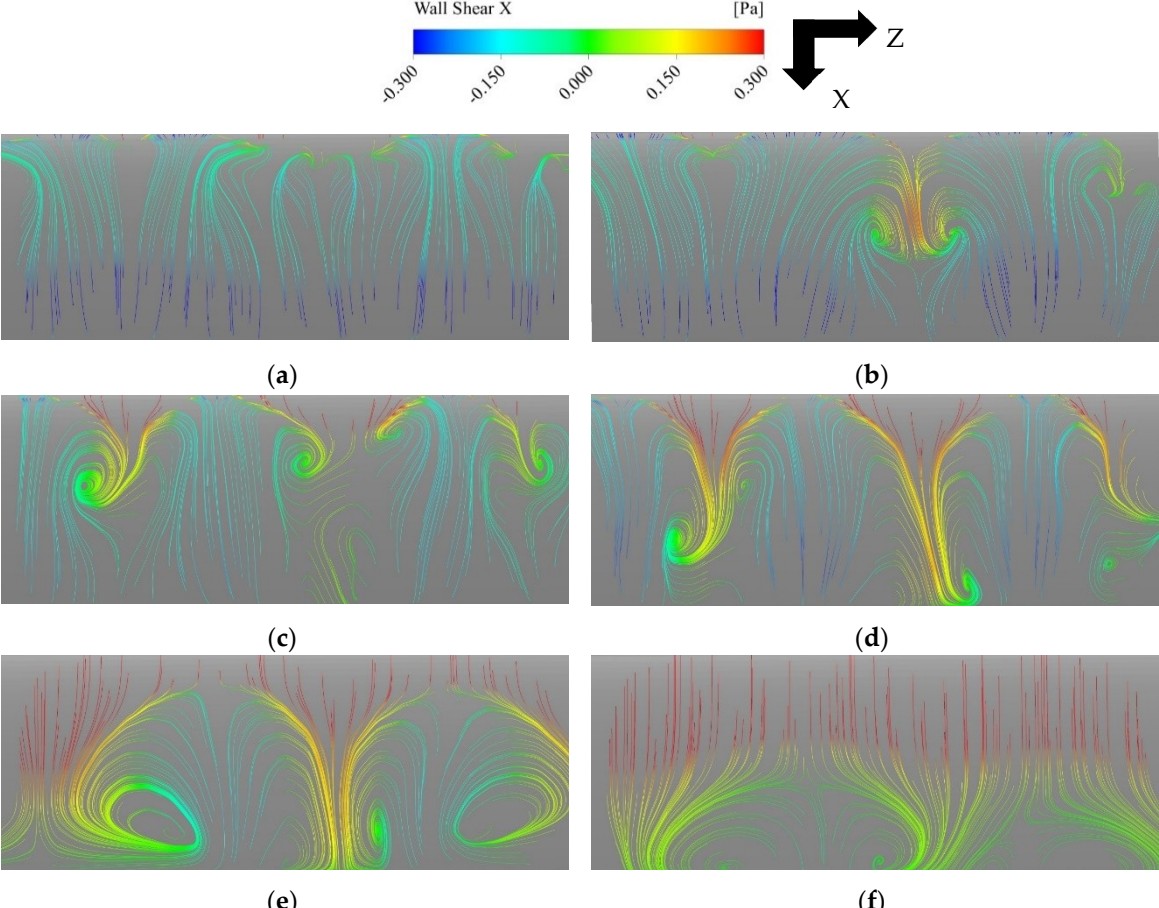

**Figure 14.** Stall cells on the aerofoil surface. Maximum AoA = 21°, $f$ = 0.01. (**a**) AoA = 18.59°↓; (**b**) AoA = 18.02°↓; (**c**) AoA = 17.39°↓; (**d**) AoA = 15.98°↓; (**e**) AoA = 14.40°↓; (**f**) AoA = 13.55°↓.

## 5. Discussion

As presented in Section 4, the $C_L$ hysteresis loop, the development of LEV, and the 3D flow structures above the aerofoil surface tend to be affected by both the reduced frequency and the pitching range (maximum AoA).

The reduced frequency has a significant impact on the hysteresis effect. A comparison of the $C_L$ hysteresis loops for the three different frequency cases with maximum AoA = 22° is shown in Figure 15. It can be seen that the stall takes place later for the cases with higher reduced frequencies, resulting in a higher maximum $C_L$. This is consistent with previous studies [1–3]. Especially for the $f$ = 0.1 case, the stall happens after the aerofoil starts pitching down and the $C_L$ does not get back to its original linear regime until around 2°. However, the linear $C_L$ curve before and after the dynamic stall is not affected substantially, which agrees with the numerical investigation of [27].

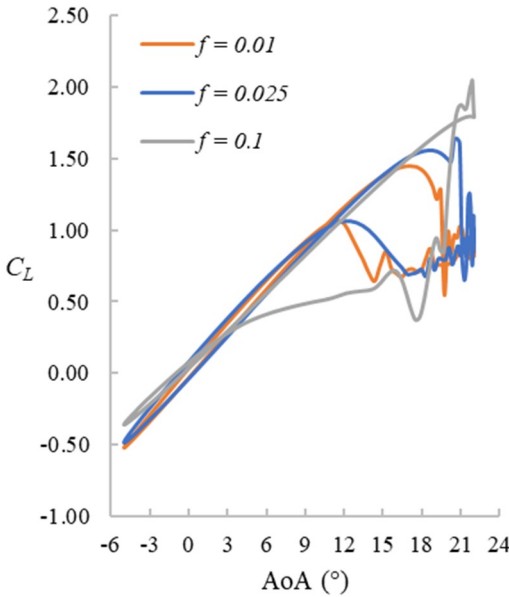

**Figure 15.** $C_L$ hysteresis loops. Maximum AoA = 22°, $f$ = 0.01, 0.025, and 0.1.

As for the development of LEV and the trailing-edge separation, cases with higher reduced frequency have stronger LEVs and weaker trailing-edge separation. As shown in Figure 7, in the $f$ = 0.1 case, a strong LEV is first developed and soon grows and takes up the whole aerofoil, whereas the trailing-edge separation bubble is pushed back by the LEV. In contrast, the streamlines in Figure 14 show a much weaker leading-edge separation and strong trailing-edge separation for the $f$ = 0.01 case. This agrees with the experimental observation of Choudhry et al. [5], who stated that low reduced frequency might fail to generate LEV. In their experimental study, the lowest reduced frequency was 0.0059, in which case the LEV was completely missing and there was no sudden increase in $C_L$.

The stall cells tend to develop more strongly at a lower reduced frequency. There is only one case in which small and weak 3D flow structures develop for $f$ = 0.1, whereas all $f$ = 0.01 cases develop clear and dominating stall cell structures on the suction side of the aerofoil. For the $f$ = 0.025 cases, the development of stall cells changes significantly with the maximum AoA. For maximum AoA = 19° to 21°, strong stall cells can develop on the aerofoil surface. However, for maximum AoA = 22° to 25°, only weak, small-sized vortex pairs are observed.

The formation of the stall cell patterns on the aerofoil surface is accompanied by 3D flow structures above the aerofoil surface. Figure 16 shows iso-surfaces of Q-criterion (at 25 s$^{-2}$, colored by streamwise vorticity) and the streamwise velocity field above the aerofoil near the trailing edge ($x$ = 0.85$c$) together with the surface flow patterns for $f$ = 0.01, maximum AoA = 24°. The figure shows clearly that the stall cells and the 3D structures above the aerofoil surface are indeed closely related to each other. The flow above the leading edge starts to form many small pairs of positive and negative streamwise vorticity regions (Figure 16a). These small 3D structures gradually evolve into larger structures like stall cells (Figure 16b,c). Finally, they merge into one large structure (Figure 16d, e) and then start to decay (Figure 16f). The wavelike separation pattern on the aerofoil surface and the arch-shaped flow structure above the surface generally resemble the Π-shaped vortices found over a pitching finite-span wing by Visbal and Gammon [22]. All these flow structures tend to move towards the trailing edge as the aerofoil is pitching down. For the cases with weak stall cells and no stall cells, the flow field above the aerofoil was found to remain 2D for the entire process of pitching.

The exact mechanism of stall cell formation is yet to be determined. Weihs and Katz [41] ascribed the phenomenon to the Crow instability [42] while Disotell and Gregory [43] claimed that stall cell formation could be related to shear layer instability. A more convincing theory was recently proposed separately by Sparlart [44] and Gross et al. [45], who analyzed the stall cell formation based on the lifting line theory. They concluded that for a massively stalled aerofoil, locally reduced circulation

could be found on the suction side, arranged alternatively with normal circulation induced by flow separation. Therefore, it understandable that such stall cells can be observed also on a slowly pitching aerofoil for the same reason. Nevertheless, as shown in the present study, it seems that LEV can suppress the stall cell as the stall cells only form during the flow reattachment stage. The relationship between stall cell formation and LEV, as well as the impact of the reduced frequency, is worth further investigation in future work.

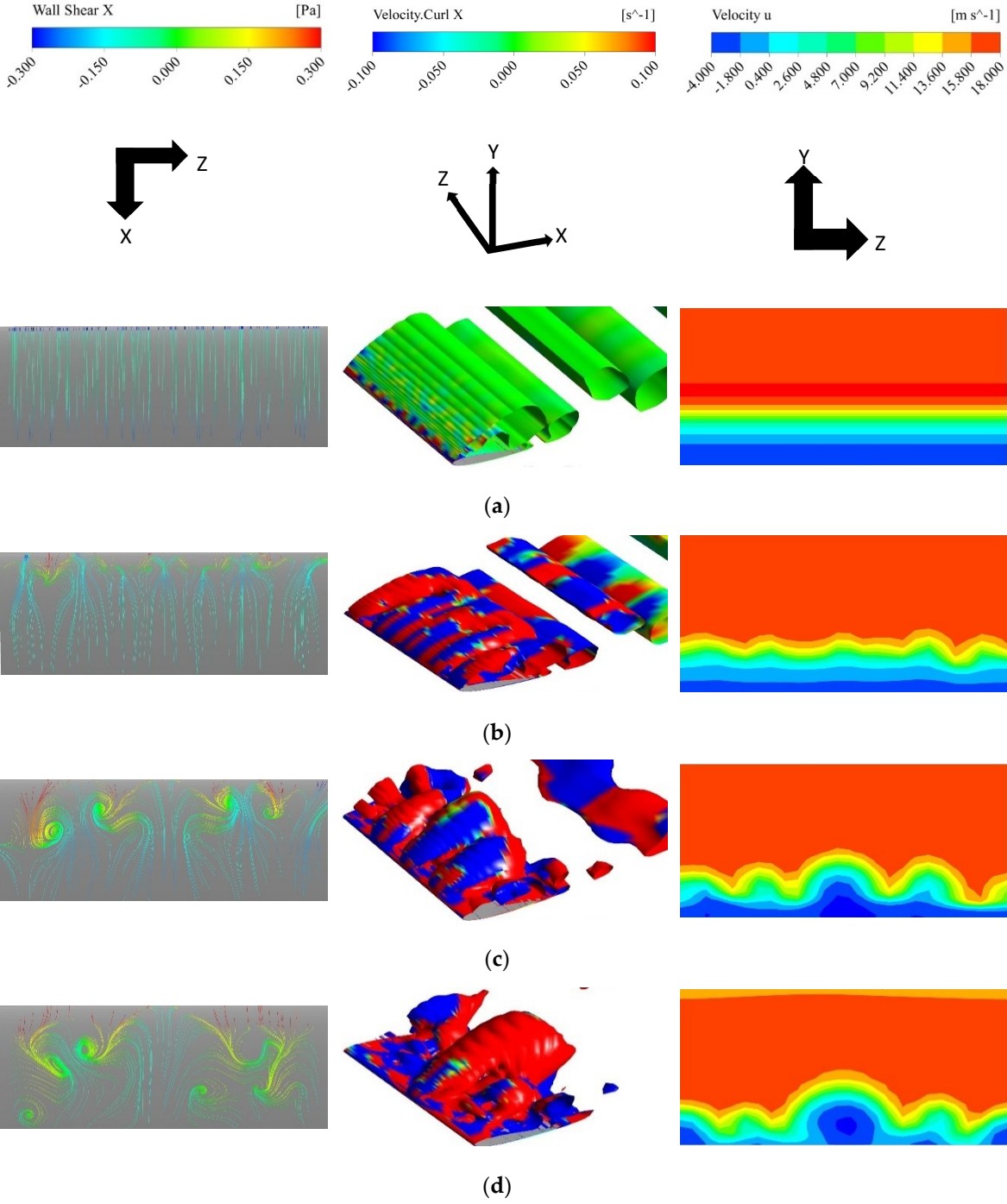

**Figure 16.** *Cont.*

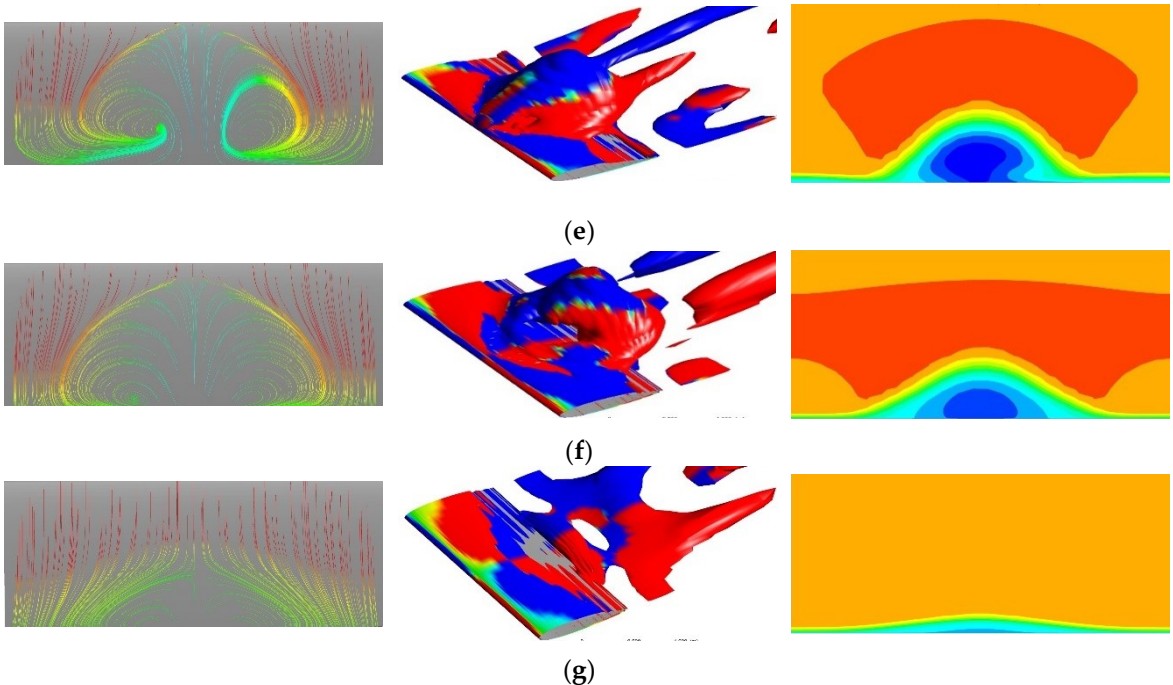

**Figure 16.** 3D flow structures on and above the aerofoil surface. $f$ = 0.01, maximum AoA = 24°.
(**a**) AoA = 19.26°↓; (**b**) AoA = 18.45°↓; (**c**) AoA = 16.69°↓; (**d**) AoA = 15.75°↓; (**e**) AoA = 14.78°↓;
(**f**) AoA = 12.76°↓; (**g**) AoA = 11.72°↓.

## 6. Conclusions

This study has investigated the impact of the reduced frequency and pitching range of a pitching NACA 0012 aerofoil on its aerodynamic behavior, the development of the LEV, and the formation of stall cells and other 3D flow structures above the aerofoil surface. A series of 3D unsteady RANS simulations have been conducted using $k$-$\omega$ SST turbulence model. Spanwise periodic boundary conditions were applied to simulate an infinitely long wing. Three different reduced frequencies were investigated: $f$ = 0.1, 0.025, and 0.01. For each case, the minimum AoA was fixed at −5°, while the maximum AoA ranged from the highest angle with which a sudden decrease of the lift did not occur for the entire pitching process (i.e., highest end of the prestall regime; this angle increases with the reduced frequency) to 25°. Therefore, in this study, the influence of the pitching range is represented by 'the maximum AoA'.

The effect of the reduced frequency and pitching range on the dynamic stall behavior and the LEV development observed in this study agree with those reported in earlier studies. For all deep-stall cases, a higher reduced frequency and a higher Max AoA both result in a higher maximum $C_L$ and a higher stall angle. The flow reattachment process is also delayed. With the highest reduced frequency ($f$ = 0.1) and a Max AoA of 21° to 24°, the stall takes place after reaching the Max AoA, in contrast to the case with the lower reduced frequencies. The LEV development is mainly influenced by the reduced frequency. For $f$ = 0.1 and 0.025, a strong LEV develops and sheds away from the aerofoil before the growth of a trailing-edge vortex. However, for $f$ = 0.01, the trailing-edge vortex keeps developing and shedding, while the LEV stays above the aerofoil and keeps expanding and shrinking.

Flow structures on the aerofoil surface have also been investigated in detail, providing new insights into the complex flow characteristics during the downstroke stage of a pitching aerofoil. Stall cells, which are often observed on a stalled static aerofoil, have been observed during the flow reattachment stage of the dynamic stall for all cases with $f$ = 0.01 and the cases with $f$ = 0.025 and Max AoA = 19°, 20°, and 21°. The strong and large stall cells travel slowly from the leading edge to the trailing edge as the aerofoil is pitching down, resulting in a slow recovery of the lift. For the cases with $f$ = 0.025 and Max AoA = 22° to 25°, however, only weak and small stall cells are

created during the downstroke stage, resulting in a faster recovery of the lift. For most cases with $f = 0.1$, the flow remains two-dimensional (except for Max AoA = 23°, where very weak, oval-shaped, three-dimensional structures are observed).

It has also been found in this study that the formation of strong stall cells on the aerofoil surface is accompanied by the development of arch-shaped 3D flow structures above the aerofoil surface. These structures are similar to the Π-shaped vortices often observed over a pitching finite aspect ratio wing [19–21]. The stall cells and the arch-shaped structures both begin with small perturbations near the trailing edge and they merge into bigger structures while moving towards the trailing edge. Since the present study is for an infinitely long wing and does not involve any wing-tip vortices, the formation of these large 3D flow structures seems to be intrinsic to low-frequency dynamic stall.

**Author Contributions:** D.L. conducted numerical modeling and data analysis, and wrote the original manuscript; T.N. supervised the project and modified the manuscript.

**Funding:** This research received no external funding.

**Conflicts of Interest:** The authors declare no conflict of interest.

## Abbreviations

| | | |
|---|---|---|
| AoA | = | angle of attack |
| AR | = | aspect ratio |
| $c$ | = | chord length |
| CFD | = | computational fluid dynamics |
| $C_D$ | = | drag coefficient |
| $C_L$ | = | lift coefficient |
| $f$ | = | reduced frequency |
| G | = | grid |
| LEV | = | leading-edge vortex |
| LSB | = | laminar separation bubble |
| $U_\infty$ | = | free-stream velocity |
| VAWT | = | vertical axis wind turbine |
| $R$ | = | computational domain radius |
| RANS | = | Reynolds-averaged Navier–Strokes |
| $Re$ | = | Reynolds number |
| SST | = | shear stress transport |
| $\alpha$ | = | pitching angle |
| $\alpha_m$ | = | mean angle |
| $\alpha_1$ | = | pitching range |
| $k$ | = | turbulent kinetic energy |
| $t$ | = | flow time |
| $x, y, z$ | = | Cartesian coordinate |
| $\Omega$ | = | angular velocity |
| $\omega$ | = | specific dissipation rate |

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
