# Peer review of "Unsteady RANS Simulations of Strong and Weak 3D Stall Cells on a 2D Pitching Aerofoil"

_fluids, doi:10.3390/fluids4010040_

Round 1
Reviewer 1 Report
Dear authors,
Please refer the attached file.

Author Response
We would like to thank the reviewer for helpful comments on our manuscript. We have provided our response to the reviewer’s comments and questions point by point in the attached Word file.

Reviewer 2 Report
There are few problems in the manuscript that needs to be corrected before its acceptance for publication. My concerns are:
1- Introduction is can be improved. Many related works such as:
https://www.sciencedirect.com/science/article/pii/S0997754609000648
https://arc.aiaa.org/doi/abs/10.2514/2.1763
are missing in the introduction. Many others that are not related are there without serving any purpose in the direction of the manuscript. References for governing equations are also missing.
2- Authors needs to explain their choice of RANS model. Also even though the choice of RANDS model is appropriate, it can not provide the full details of flow as it is seen in DNS. Please see:
https://www.sciencedirect.com/science/article/pii/S0960148116311326
3- Many parameters are not defined in the text and therefore, it is very difficult to follow the equations and the text. Please add a nomenclature.
4- In fig.1 what is the edge limit for the inlet and outlet? Mesh Convergence is also missing! Please show the convergence of your mesh in a separate figure/table.
5- Are the structure seen in figure 15 are the Gortler type of instability? please see:
https://www.sciencedirect.com/science/article/abs/pii/S0045793018303335
https://aip.scitation.org/doi/abs/10.1063/1.865531
6- Quality of the presented figures can be improved.
Author Response
We would like to thank the reviewer for the inspiring and constructive comments on our manuscript. We have provided our response to the reviewer’s comments and questions point by point in the attached Word file.

Round 2
Reviewer 1 Report
Dear authors,
appropriate responses are given to all the comments.
Reviewer 2 Report
authors sufficiently answered my concerns.